# Potential Role of Lycopene in the Prevention of Postmenopausal Bone Loss: Evidence from Molecular to Clinical Studies

**DOI:** 10.3390/ijms21197119

**Published:** 2020-09-27

**Authors:** Umani S. Walallawita, Frances M. Wolber, Ayelet Ziv-Gal, Marlena C. Kruger, Julian A. Heyes

**Affiliations:** 1School of Food and Advanced Technology, Massey University, Palmerston North 4442, New Zealand; u.walallawita@massey.ac.nz (U.S.W.); F.M.Wolber@massey.ac.nz (F.M.W.); 2College of Veterinary Medicine, University of Illinois at Urbana-Champaign, Urbana, IL 61802, USA; zivgal1@illinois.edu; 3School of Health Sciences, Massey University, Palmerston North 4442, New Zealand; m.c.kruger@massey.ac.nz

**Keywords:** bone, osteoporosis, lycopene, tomato, postmenopause

## Abstract

Osteoporosis is a metabolic bone disease characterized by reduced bone mineral density, which affects the quality of life of the aging population. Furthermore, disruption of bone microarchitecture and the alteration of non-collagenous protein in bones lead to higher fracture risk. This is most common in postmenopausal women. Certain medications are being used for the treatment of osteoporosis; however, these may be accompanied by undesirable side effects. Phytochemicals from fruits and vegetables are a source of micronutrients for the maintenance of bone health. Among them, lycopene has recently been shown to have a potential protective effect against bone loss. Lycopene is a lipid-soluble carotenoid that exists in both all-*trans* and *cis*-configurations in nature. Tomato and tomato products are rich sources of lycopene. Several human epidemiological studies, supplemented by in vivo and in vitro studies, have shown decreased bone loss following the consumption of lycopene/tomato. However, there are still limited studies that have evaluated the effect of lycopene on the prevention of bone loss in postmenopausal women. Therefore, the aim of this review is to summarize the relevant literature on the potential impact of lycopene on postmenopausal bone loss with molecular and clinical evidence, including an overview of bone biology and the pathophysiology of osteoporosis.

## 1. Introduction

Osteoporosis is an age-related health problem which reduces the quality of life. It is estimated that over 200 million people suffer from osteoporosis worldwide [1]. Approximately 30% of postmenopausal women in Europe and the United States are affected by osteoporosis [2]. Based on the projected estimations, there will be a 32% increase in the incidence of osteoporosis and low bone mass in older adults (aged ≥ 50) by 2030 [3]. Moreover, the cost of treatments for osteoporosis is rapidly increasing annually as the total number of fractures increases. Developing countries especially may face a huge economic burden due to more fragility fractures occurring with higher life expectancies and a growing elderly population [4,5]. The projected estimate of the increase in the incidence of hip fractures in women and men worldwide is 240% and 310%, respectively, by 2050.

The most prominent types of fractures occur at the sites of the hip, spine vertebrae, and distal forearm [4]. Complications of osteoporosis, such as hip fractures, lead to a four-times-higher mortality rate in the adult population worldwide [6]. In general, one in three women (50 years or older) may have osteoporotic fractures [7]. Osteoporosis can be categorized into two types: primary osteoporosis and secondary osteoporosis [8]. Primary osteoporosis can be further divided into postmenopausal osteoporosis (type 1) and senile osteoporosis (type 2) [9]. Postmenopausal osteoporosis is primarily caused by estrogen deficiency, which occurs in women after menopause, while senile osteoporosis gradually develops with aging (>70 years old) in both sexes [10]. Secondary osteoporosis generally has a definable etiology such as malnutrition, chronic disease, endocrine dysfunction, medication side effects, or metastatic or hematological malignancy [8].

Diets rich in phytochemicals, particularly carotenoids from fruits and vegetables, have been found to be effective for the maintenance of bone mineral status [11,12]. Among them, the potential protective effects of lycopene against bone loss have been recently documented [13,14]. Lycopene is an acyclic carotenoid, containing 11 conjugated double bonds in the all-*trans*-isomeric form or various *cis*-configurations [15]. Compared to all-*trans*, *cis*-isomers are considered to be more bioavailable because they are highly soluble in lipid micelles, readily taken up by intestinal cells, less likely to crystalize, and they also preferentially bind to chylomicrons [16]. Thus, *cis*-isomers are more easily transported within cells, across the plasma membrane, and into the tissue matrix [16,17].

Tomatoes and tomato-based products are rich sources of lycopene and represent more than 80% of human dietary sources containing lycopene. Depending on the variety of tomato, the average lycopene content lies between 0.7–20 mg/100 g fresh weight [18]. However, dietary sources of lycopene are primarily found in the all-*trans*-isomeric form. For example, red tomatoes contain almost 90% of their total lycopene in the all-*trans*-form [15], yet *cis*-lycopene concentrations in body tissues are higher than those of other lycopene isomers [18,19]. This could be due to isomerization occurring during food processing or post consumption during digestion and absorption. Although lycopene has beneficial effects on human health, there is a gap between dietary intake and the amount available for biological action in the body. Therefore, various food processing techniques (e.g., heat treatment) are used to improve the bio-accessibility of all-*trans*-lycopene. These processing techniques can disrupt the cell wall and release lycopene out of the cells. Additionally, since lycopene is fat-soluble, the incorporation of fat into the meal or red tomato product increases the bioavailability of lycopene. There are other natural sources of more bioavailable lycopene, such as orange heirloom tomatoes that contain >90% of lycopene in the *cis*-isomeric form [20,21]. In these tomatoes, the carotenoid isomerase enzyme, which converts *cis*-lycopene to all-*trans*, is in a nonfunctional form; thus, they retain considerably higher amounts of *cis*-isomers and may be a better source of dietary lycopene for humans.

There is an emerging interest among researchers to study lycopene for the prevention of postmenopausal osteoporosis and to explore ways to improve lycopene bioavailability from natural sources such as tomatoes. The purpose of this review is to discuss the literature regarding the impact of lycopene on bone metabolism, including an investigation into natural sources of more bioavailable lycopene.

## 2. Bone Biology, Modelling, and Remodeling

Bone is a specialized connective tissue that is responsible for the framework of the body. Primarily, the skeleton provides support for the body and assists its movements. Bones also act as a major mineral reservoir, carrying 99% of calcium, 85% of phosphorus, and 65% of magnesium body stores, and they are the main repository of growth factors and cytokines [8]. Moreover, bone plays an important role in acid-base balance and hormonal functions related to phosphate metabolism, blood glucose, and fat deposition in the body [8]. Bone also helps in the detoxification of heavy metals and other waste materials by removing them from blood circulation [8,22,23]. Bone generally comprises three components: organic matrix, inorganic salts, and water. Approximately 90% of the organic matrix consists of collagenous protein, non-collagenous protein, and growth factors. The inorganic matrix contains mainly calcium and phosphorus in the form of hydroxyapatite crystals [24]. Depending on the degree of porosity, bones are categorized as cortical bone (compact) or trabecular bone (cancellous). Cortical bone is denser, while trabecular bone is more porous. The porosity of cortical bone is approximately 3–5% and may increase with age [23,25]. Cortical bones represent 80% of the adult skeleton. Trabecular bone is more metabolically active than cortical bone. This difference occurs due to the higher surface area of trabecular bone compared to cortical bone [23].

Bone tissue contains four different cell types: osteoblasts, osteoclasts, osteocytes, and bone lining cells. Osteoblasts originate from mesenchymal stem cells and are responsible for bone formation [26]. Osteoclasts are derived from mononuclear hematopoietic stem cells and are responsible for bone resorption [27]. The majority of mature osteoblasts may undergo apoptosis, while a minority re-differentiate into osteocytes or lining cells [28]. Approximately 90–95% of bone cells are osteocytes, which have a long life span of nearly 25 years [24,29]. Bone marrow, found in the bone cavity, consists of two types of cells: hematopoietic and stromal. Hematopoietic stem cells produce osteoclasts, immune cells, platelets, and red blood cells, while mesenchymal stem cells produce osteoblasts, cartilage, and adipocytes [27,29].

Bone tissue undergoes two major physiological processes: modeling and remodeling. Bone modeling is characterized by a change in the shape of the bone as a result of physiological influences or mechanical forces [30]. For example, bone modeling widens the bones with aging [8]. Moreover, bone modeling is upregulated in hypoparathyroidism, chronic kidney disease (CKD), and medical treatments containing anabolic agents [31]. During bone modeling, the changes in the shape of bones are regulated by independent actions of osteoblasts and osteoclasts [32].

In contrast, bone remodeling occurs throughout life and is responsible for the removal of older bone and its replacement with new bone structure [8,33,34]. Bone remodeling involves a sequence of cellular activities that occur within a specialized multicellular unit [35]. The bone multicellular unit is predominantly comprised of osteoclasts, osteoblasts, and osteocytes [24]. There are 3–4 million basic multicellular units (BMU) produced each year, and approximately one million among them actively participate in the bone remodeling process [36]. Bone remodeling has four major stages: activation, resorption, reversal, and formation [35]. In the first step, osteoblastic stromal cells or lining cells are activated via lining cell retraction and endosteal membrane digestion by collagenase. Following this, osteoclasts initiate bone resorption by dissolving the mineral matrix [8,35]. At the end of the resorption phase, mononuclear cells such as monocytes, osteocytes, and preosteoblasts are found in resorption cavities [8]. During the reversal phase, a cement lining rich in mucopolysaccharides is deposited between old and new bone, as well as signaling molecules that can activate osteoblast precursors. Therefore, the reversal phase is considered a transitional phase between resorption and formation of new bone [35]. Lastly, a new organic matrix is produced by osteoblasts, which eventually mineralizes into new bone [32]. The resorption and reversal phases last for 2 weeks and 4–5 weeks, respectively. The formation phase is the longest and lasts approximately 4–6 months until the new bone is completely formed [30,37,38].

Approximately 90% of cortical bone is calcified; thus, it contains a low surface area to volume ratio. This leads to a slower rate of remodeling in cortical bone compared to trabecular bone [23]. Approximately 25% of the body’s trabecular bone is remodeled each year compared to only 2.5% of cortical bone [23]. Bone remodeling is regulated by various systemic and local factors. Genetics, mechanical factors, vascular factors, nutrition, and hormones are considered systemic regulators, while growth factors, matrix proteins, and cytokines act as local regulators [8,24,35,37,39]. At menopause, bone remodeling increases and continues at a higher rate for 5–10 years due to the decrease in levels of estrogen [31].

## 3. Postmenopausal Osteoporosis: A Silent Disease

Postmenopausal osteoporosis is a common metabolic disease among older women (≥50 years). It is characterized by reduced bone mineral density along with disruption of bone microarchitecture and alteration of non-collagenous protein in bone, which together lead to higher fracture risk [8]. This disease reduces women’s overall quality of life by significantly increasing their rates of morbidity, disability, and mortality. Osteoporotic bone tends to be more fragile and easily fractured [40]. In general, one in eight people will experience a second fracture within a year after their first osteoporotic fracture [41].

Bone mineral density (BMD), bone mineral content (BMC), and the quantity and quality of bone are the distinguishing factors between people with osteoporotic versus healthy bones [42]. The risk of osteoporosis is primarily evaluated through bone mineral density, which is predominantly measured by dual-energy X-ray absorptiometry (DEXA) scanning [38]. Other methods include magnetic resonance imaging (MRI), ultrasound, and microcomputed tomography [42,43]. Bone density is a quantitative measure, but its limitation is that it does not measure bone quality [44]. Therefore, measurements of bone turnover markers as proxies for bone formation and bone resorption are used to evaluate the quality of bone in tandem with DEXA scanning [38].

For the diagnosis of osteoporosis, BMD values are converted into a T-score [2]. The T-score is calculated by dividing the difference between a female patient’s BMD and the mean BMD of young, healthy women by the standard deviation of the reference population [2]. A T-score between −1 and +1 SD (standard deviation) is considered to be normal BMD. A T-score between −1 and −2.5 SD is categorized as osteopenia. A T-score of −2.5 SD is considered to be osteoporotic [27,40,43,45,46]. An imbalance between bone resorption and formation initially leads to osteopenia, which is characterized by low mineralization and is likely to further progress to osteoporosis [47]. The most common pharmacological treatments currently being used for osteopenia and osteoporosis are bisphosphonates, denosumab, anabolic agents, and hormone replacement therapy [41]. Most of these treatments have been shown to increase bone mass by up to 10% over 3–5 years [48]. Postmenopausal women are advised to take these medications with calcium and vitamin D supplements to increase their effectiveness [49].

### 3.1. Risk Factors of Postmenopausal Osteoporosis

Risk factors for osteoporosis can be primarily categorized as modifiable and nonmodifiable (Table 1). Heredity is the major nonmodifiable factor of osteoporosis, and children of parents with osteoporosis and fractures are themselves more prone to develop osteoporosis [50]. Osteoporosis is a polygenic disease that involves several genes. In rare instances, osteoporosis can be inherited due to mutations in single genes. Mutations of two genes of type 1 collagen (COL1A1 and COL1A2) are responsible for the dominant osteoporotic disease called “osteogenesis imperfecta”, which is characterized by low bone mass and increased bone fragility [51,52,53,54]. Osteoporosis inheritance has also been linked with inactivating mutations in the aromatase (CYP19A1) and estrogen receptor alpha genes (ERα) [55,56].

Estrogen deficiency is the primary risk factor of postmenopausal osteoporosis; other contributors besides the genetics mentioned above include modifiable factors such as nutrition, certain medications, and lifestyle (Table 1) [10,74]. Bone cells (osteoblast, osteoclast, and osteocytes) contain estrogen receptors on their surface [75]. Stimulation of estrogen receptors, particularly on osteoblasts, may inhibit the activation of osteoclasts and thus reduce bone resorption and protect bones from osteoporosis [76]. Oxidative-stress-generating factors such as poor nutrition, low antioxidants in the body, smoking, alcohol intake, and excessive caffeine intake can be modified through lifestyle changes [6,40]. For example, a diet low in calcium may induce secretion of parathyroid hormone, which activates osteoclasts and bone resorption [76].

Interestingly, some cross-sectional studies have found an inverse relationship between sleep duration and BMD in elderly women [77,78]. It has been hypothesized that a shorter waketime reduces the daily mechanical loading that induces bone remodeling, and thus reduces BMD [78]. Lower melatonin levels due to reduced light exposure may lead to fewer interactions between estrogen and its receptors and, thus, negatively impact BMD [79]. However, other studies report either positive or null relationships between sleep duration and BMD in postmenopausal women [80,81,82,83]. The conflicting evidence in this area may be due to confounding study participant factors that are not uniformly controlled for across the studies, such as age, body composition, diet, and nighttime-only sleep duration versus inclusion of daytime naps.

### 3.2. Pathophysiology of Postmenopausal Osteoporosis

The occurrence of postmenopausal osteoporosis is dependent mainly on body estrogen levels. Estrogen regulates bone turnover either by directly interfering with osteocytes and osteoclasts or by regulating T-lymphocyte function and the formation of osteoblasts [84]. Estrogen has both skeletal and non-skeletal functions, and, due to the former deficiencies, can cause bone-related diseases (Figure 1). Women are at higher risk 3–5 years after the onset of menopause [85]. The mechanism by which estrogen deficiency causes postmenopausal osteoporosis is complex. Estrogen can influence bone remodeling through inhibiting cell differentiation and increasing osteoclast apoptosis [85].

Molecular markers of bone metabolism are widely used to assess bone-related disorders. These markers include enzymes and nonenzymatic peptides produced by bones. Bone formation or resorption markers correlate with the metabolic phase in which they are produced [38]. Short-term estrogen deficiency (3 weeks) is associated with low levels of bone formation markers in early postmenopausal women [87]. Interestingly, studies have shown that long-term estrogen deficiency increased both bone resorption and bone formation markers in postmenopausal women, suggestive of enhanced bone turnover with increased net bone loss [88,89,90,91]. Estrogen deficiency increases renal calcium excretion while decreasing intestinal calcium absorption [10], and the resultant fall in calcium levels can activate various bone resorption mechanisms that include PTH, osteocalcin, OPG, and the RANK/RANKL system [91,92,93,94,95]. These bone resorption markers are, therefore, found in the blood in higher concentrations in osteoporosis. Conversely, osteocalcin, which is secreted by osteoclasts, directly binds calcium and enables bone mineralization by increasing hydroxyapatite absorption and, thus, is a marker of bone formation [96]. However, insufficient calcium and phosphorous stores in osteoporotic women reduce hydroxyapatite crystal formation, leaving more osteocalcin free to circulate in the blood [93,94]. The molecular mechanisms responsible for these complex changes are not yet fully elucidated [88].

As mentioned earlier, oxidative stress can be coupled with osteoporosis. Oxidative stress occurs when there is an imbalance between the production of reactive oxygen species (ROS) and their neutralization by antioxidants. Reactive oxygen species are formed as a result of cellular respiration, enzymatic activities in mitochondria, and cellular responses to cytokines induced by external stimuli [76]. Reactive oxygen species include both highly reactive oxygen-containing molecules and free radicals such as hydroxyl (OH), superoxide (O_2_^−^), and hydrogen peroxides (H_2_O_2_) [40]. Free radicals can oxidize lipids and proteins, thus causing cell damage and altered function [97,98]. Reactive oxygen species suppress differentiation and proliferation of osteoblasts and are significantly involved in osteoclast differentiation and bone resorption [99,100]. Menopause increases oxidative stress; thus, the oxidized microenvironment produced by ROS plays a major role in causing postmenopausal osteoporosis [85,99,101]. Antioxidants are directly involved in the scavenging process of ROS. A lack of antioxidants may increase proinflammatory cytokines, especially tumor necrosis factor (TNF-α), and thereby induce bone loss [98].

There are two cytokines primarily responsible for osteoclastogenesis: macrophage colony-stimulating factor (M-CSF) and receptor activator of nuclear factor kappa B ligand (RANKL), which are produced by bone marrow stromal cells and osteoblasts during bone remodeling [31]. The RANK (receptor activator of nuclear factor kappa B), RANKL, and OPG system has been identified as a primary regulator of the bone remodeling process. Osteoprotegerin (OPG), which is produced by osteoblasts, is considered to be a decoy receptor for RANKL. RANKL binds to its receptor RANK on osteoclast precursors in the presence of M-CSF. Upon this, osteoclast precursors differentiate and combine to form multinucleated osteoclasts, which can start bone resorption [85]. Due to insufficient estrogen levels in postmenopausal women, OPG is downregulated and RANKL activity upregulated, thereby increasing osteoclastogenesis [10].

## 4. Carotenoid Lycopene: Chemistry and its Isomers

To date, more than 700 carotenoids [102] have been identified, of which 40–50 are present in the human diet in fruits and vegetables [103]. However, only 20 have been found in human tissues or blood [104,105]. There are two classes of carotenoids: nonoxygenated carotenoids and oxygenated carotenoids. Nonoxygenated carotenoids are unsaturated hydrocarbons such as lycopene, α-carotene, β-carotene, γ-carotene, and ζ-carotene, whereas oxygenated carotenoids are the xanthophylls [18].

The major carotenoids found in the human body and human diet are β-carotene, α-carotene, α-cryptoxanthin, lutein, zeaxanthin, and lycopene [106,107]. Carotenoids are localized within chloroplasts and chromoplasts in plant cells. In chloroplasts, the carotenoids are found in association with proteins, whereas chromoplasts contain a crystalline form of carotenoids [108]. Lycopene, a member of the carotenoid pigment family, is responsible for the specific red color in many fruits and vegetables, best typified by fresh tomatoes and tomato products [107]. It is a lipid-soluble antioxidant produced by plants and some microorganisms. Unlike β-carotene, lycopene does not contain a terminal β-ionone ring and thus does not have provitamin A activity. Lycopene is an acyclic carotenoid containing 11 conjugated double-bonds in its all-*trans*-isomeric form or in various *cis*-configurations (Figure 2) [15]. Having 11 conjugated double-bonds, lycopene is theoretically assumed to have 2048 possible *cis–trans*-conformations in nature, but only a few have been identified so far. Due to its molecular structure, only certain *cis*-isomers exist, predominately 5-*cis*, 9-*cis*, 13-*cis*, and 15-*cis*. The most stable isomeric form is 5-*cis*, followed by all-*trans*, 9-*cis*, 13-cis, 15-*cis*, 7-*cis*, and, finally, 11-*cis* as the least stable [104].

Phytochemicals from fruits and vegetables are reported to aid in the maintenance of bone metabolism. In particular, carotenoids such as α-carotene, β-carotene, canthaxanthin, and lycopene have demonstrated beneficial effects on skeletal health; there is a clear positive association between lycopene intake and reduced bone loss in humans [109]. Recently, an inverse relationship between hip fracture risk and consumption of carotenoids from fruits and vegetables was reported in men aged 45–74 [110]. Lycopene may suppress the formation of preosteoclasts from osteoprogenitor cells, thereby disrupting the osteoclast formation pathway [5].

### Lycopene Bioavailability, Absorption, and Metabolism

The bioavailability of ingested lycopene is dependent on the dose of lycopene consumed, linkages between molecules in the food matrix, incorporation of fats, level of dietary fiber, interactions of lycopene with other carotenoids, and genetic factors [18,111]. Moreover, the bioavailability of lycopene differs depending on the isomeric form. In fact, *cis*-isomers of lycopene are estimated to be 8.5 times more bioavailable than all-*trans* lycopene [16]. The higher bioavailability of *cis*-isomers compared to all-*trans* is mostly due to the former’s increased solubility in mixed micelles [112]. Compared to all-*trans*, *cis*-isomers are less likely to crystalize, highly soluble in oil, preferentially micellarized, readily taken up by intestinal cells, and easily transported within cells as well as across plasma and tissue matrices [16,17]. It is suggested that due to the polar nature and kinked forms of *cis*-isomers, they are less prone to crystallize [18,113]. Moreover, their preferential solubility is likely due to the smaller chain length of *cis*-isomers based on the bending of their structures, which may not be found in *trans*-lycopene isomers [114]. Lastly, thermal processing and mixing with oil can further increase the bioavailability of *cis*-lycopene present in tomatoes [18].

Interestingly, there are some natural sources, such as orange heirloom tomatoes, that contain more bioavailable *cis*-lycopene [15,20]. Almost 90% of the lycopene in orange heirloom tomatoes is in the *cis*-form compared to 90% of the all-*trans*-isomeric form of lycopene in red tomatoes [15]. This is due to a mutation known as tangerine in the carotenoid *cis-trans*-isomerase gene [115]. The *cis*-form gives a deep orange color to these tomatoes [20,116]. As the tangerine mutant lacks the ability to convert poly-*cis*-lycopene into all-*trans*, orange heirloom tomatoes predominantly accumulate tetra-*cis*-lycopene along with other precursors of lycopene such as phytoene, phytofluene, ζ-carotene, and neurosporene [20].

Being lipid-soluble, lycopene is absorbed into the body following the same pathway as fats. The foremost step of lycopene absorption is the breakdown of the food matrix and the release of carotenoids into the gastrointestinal lumen. Mechanical alteration of the food structure by cooking and other methods of food processing may improve carotenoid release from the food matrix [117]. Lycopene then enters intestinal mucosal cells through the formation of bile acid micelles [118]. However, bile production depends on the amount of fat present in the diet; therefore, it is necessary to incorporate fat with the lycopene-containing food in order to increase its solubility [119]. In general, lycopene absorption from dietary sources ranges from 10–30% [104], and, according to previous studies, a minimum 5–10 g of fat in a meal is required to ensure better absorption of carotenoids [120]. Once lycopene enters the enterocytes, it is either cleaved by β-carotene−9′,10′-oxygenase (BCO2) to produce lycopenoids or incorporated into chylomicrons and secreted into lymphatic and blood circulation [117,120]. Absorbed lycopene can either accumulate in the liver or be packed into VLDL and HDL and thereby re-enter the blood. Via blood circulation, lycopene is deposited in extrahepatic organs, mainly the adrenal glands, adipose tissue, prostate, and testes [121]. Non-absorbed lycopene and the excess metabolic products are excreted from the body in urine and feces (Figure 3).

Lycopene concentrations in body tissues are higher than those of other carotenoids [18,19,119]. Half of the carotenoids in human serum are lycopene, and, among them, *cis*-isomers account for 58–73% of total lycopene in serum [16,122,123]. In the human body, lycopene is present at 1 nmol/g in adipose tissues and is found in higher concentrations of up to 20 nmol/g in testes, adrenal, and prostate glands [124]. Interestingly, neither lycopene nor other carotenoids have been found in brain stem tissues [125]. In addition, lycopene appears to have a long half-life; a recent study indicated that lycopene and its metabolites could be detected in the skin of humans up to 40 days after consumption [126].

The distribution of lycopene isomers is similar between plasma and tissues. Regardless of the isomeric forms of lycopene consumed, plasma and other tissues contain more *cis*-isomers, mainly the 5-*cis*-form [119,122]. Lycopene metabolites, or lycopenoids, can be the products of lycopene metabolism and oxidation. Kopec et al., were the first to identify the series of lycopenoids present in human blood, but they are found in plasma in only negligible amounts [127]. Similarly, lycopene metabolites can be found in some foods, but at 1000-fold lower concentrations than that of lycopene itself [127]. Some studies suggest that metabolites of lycopene may play a role in the biological activities of lycopene. However, only a limited number of studies have investigated the role of lycopene metabolites in vivo [128,129,130,131,132].

## 5. Evidence of the Effect of Lycopene on Bone Health

Recently, the effect of lycopene on bone health has received additional attention from researchers [137]. The beneficial effects of lycopene on bone health have been studied using animal models, cell cultures, and epidemiological/clinical studies, as described below and in Table 2, Table 3 and Table 4, in the context of postmenopausal osteoporosis.

### 5.1. Epidemiological and Clinical Studies

Human epidemiological studies have investigated the specific effects of tomato/lycopene on bone health, and the majority have shown a positive correlation between tomato/lycopene consumption and the prevention of bone loss (Table 2). The Framingham Osteoporosis study evaluated associations between total and isolated carotenoids with BMD in older adults (~75 years old) [138]. An inverse relationship between lycopene levels and four-year bone loss in the lumbar spine in older women (~75 years old) was observed [138], and a follow-up study reported a protective effect of lycopene against hip fractures [139]. Mackinnon et al. reported a notable increase in a clinically relevant bone resorption marker, the crosslinked N-telopeptide of type 1 bone biomarker (NTx), as well as oxidative stress markers in postmenopausal women after one-month restriction of lycopene in the diet [140]. This also led to a drastic reduction in serum lycopene along with other carotenoids such as α-carotene, β-carotene, lutein, and zeaxanthin [140]. Similarly, 30 mg/d lycopene supplementation in postmenopausal women in either juice or capsule form for four months decreased serum NTx level [141]. An epidemiological study in premenopausal women, which evaluated the effect of dietary carotenoids on bone mineral status, showed a positive correlation between lycopene intake and total body BMC and BMD [12]. Another study revealed lower levels of serum lycopene in postmenopausal women with osteoporosis compared to non-osteoporotic women [142].

### 5.2. Animal Trials

Ovariectomy (OVX) is the most widely used surgical technique for the induction of osteoporosis in rodents and other animal models to mimic the hormonal and skeletal status of postmenopausal women [146]. Along with ovariectomy, most trials incorporate “sham”-operated animals as controls, which undergo surgery without the ovaries being removed [42]. Female rats are considered an excellent animal model for postmenopausal osteoporosis, but the age of the animals and site selection for harvesting the bones must be defined with care. Other experimental protocols related to bone loss have used interventions such as hormonal, dietary, and immobilization in rats, which had more variable effects on the rate of bone loss [147]. Ovariectomized rats fed a low calcium diet demonstrated rapid bone loss, supporting the use of this model as a gold standard for the evaluation of drugs for the treatment of osteoporosis [148,149]. Higher serum levels of osteocalcin, a biomarker of bone turnover, were measured in ovariectomized rats fed a low calcium diet compared to sham-operated rats [38,149]. Ovariectomized rats are considered most suitable for the evaluation of preventative agents for postmenopausal osteoporosis, although not for evaluation of treatment of osteoporosis over a long period of time [42,150]. This is because the rate of bone turnover in OVX rats become similar to their sham counterparts in studies >12 months, and also because the exact parameters for expected increases in the skeletal size of trabecular and cortical bone in the rat with long term OVX are not yet established [150].

Lycopene has been shown in multiple studies to dose-dependently increase BMD in ovariectomized rats [151]. Similarly, lycopene reduced bone fragility in ovariectomized rats and improved femoral bone energy, as assessed using a mechanical breaking test [14]. More recently, lycopene supplementation in ovariectomized rats was found to significantly alter levels of biomarkers of bone turnover in blood and urine, reducing bone resorption and increasing osteoblast activity. Simultaneously, lycopene treatment increased the enzyme action of glutathione peroxidase, catalase, and superoxide dismutase, and downregulated oxidative stress [99]. Taken together, these animal studies suggest that lycopene has bone-protective benefits (Table 3).

### 5.3. Bone Cell Culture Studies

Multiple in vitro studies have demonstrated that lycopene and other carotenoids directly affect both osteoclasts and osteoblasts (Table 4). Ishimi and coworkers reported that osteoclast-like cell formation induced by 1α, 25(OH) 2 D3 (Calcitriol), IL-1β, and parathyroid hormone was inhibited by retinoic acid and carotenoids, including β -carotene, canthaxanthin, and lycopene, with retinoic acid being most effective [154]. Park et al., also found retinoic acid to have the highest activity in inducing differentiation of the osteoblastic cell line MC3T3-E1, with retinol, β -carotene, lycopene, and canthaxanthin showing lesser but similar effectiveness [155]. Lycopene inhibited osteoclast formation and bone resorption by rat bone marrow cells in a model of parathyroid hormone-induced osteoclastogenesis [34]. More recently, human osteoclast and osteoblast precursor cells, treated with 500 nM lycopene for 21 days, demonstrated reduced osteoclastogenesis while increasing osteoblastogenesis [156]. Lack of nuclear factor kappa B (NFκB) may lead to a reduction in osteoclast precursors differentiating into mature osteoclasts, thereby reducing bone loss. Interestingly, some studies suggest that derivatives of carotenoids can downregulate the activity of NFκB activity, a regulator of cytokine expression [8,157,158], suggesting NFκB modulation is a candidate pathway for lycopene’s protective effects against osteoporosis. Taken together, evidence from in vitro studies suggest that lycopene has beneficial effects on bone health via molecular mechanisms that are summarized in Figure 4.

## 6. Conclusions

Postmenopausal bone loss is a public health issue in the aging population. Medications can prevent and treat bone loss in some patients but may have negative side effects. Phytochemicals present in fruits and vegetables play a major role in noncommunicable disease management. As summarized in this review, lycopene has a protective effect against bone loss; this has been demonstrated in in-vitro studies, in animal models of osteoporosis, and human clinical studies. The epidemiological and clinical studies discussed in this review have demonstrated that lycopene intake (≥30 mg/day) is effective in reducing bone resorption markers in postmenopausal women [140,141]. Tomatoes are a major source of dietary lycopene; however, due to the relatively low bioavailability of all-*trans*-lycopene isomers found in red tomatoes, several techniques are used to increase bioavailability, such as adding oils or making cooked sauces and pastes. Other sources of more bioavailable lycopene can be found in nature. For example, orange heirloom tomatoes contain >90% of lycopene in a *cis*-isomeric form, which is estimated to be 8.5 times more bioavailable than all-*trans*-lycopene [16]. Further studies are warranted to compare the relative benefits of these tomato varieties and lycopene isomers in protecting against bone loss.

## Figures and Tables

**Figure 1 ijms-21-07119-f001:**
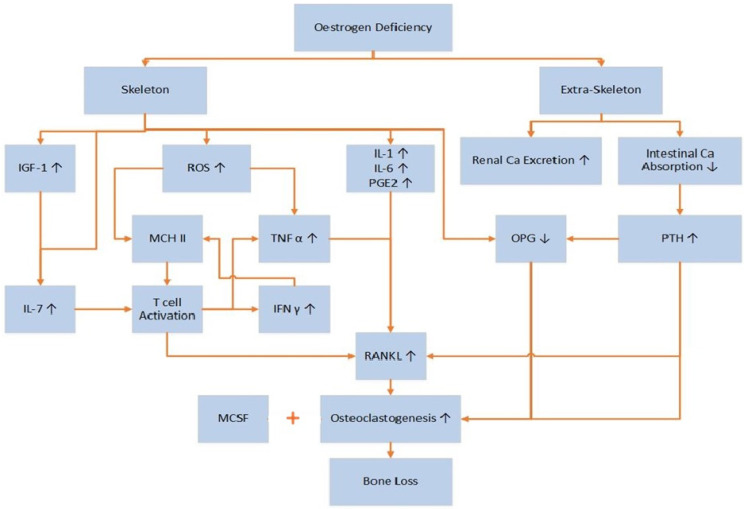
Occurrence of bone loss through estrogen deficiency (reference from [10,86]). Estrogen deficiency increases the production of IL-7 directly and via increased production of IGF-1. IL-7 activates T-cells to produce IFN-γ and TNF-α. Reactive oxygen species (ROS), along with IFN-γ, upregulate MHC II, located in antigen-presenting cells that may further activate T-cells. Activated T-cells produce RANKL and TNF-α. Other cytokines, IL-1, IL-6, and PGE2, also increase the production of RANKL. Decreased osteoprotegerin (OPG) due to insufficient estrogen directly influences osteoclastogenesis. Beyond the skeletal activities, estrogen deficiency may increase renal calcium excretion while decreasing intestinal calcium absorption. This stimulates the parathyroid glands to produce PTH, which can reduce the production of OPG and increase the production of RANKL and, therefore, increase bone resorption. All these actions together are involved in postmenopausal bone loss.

**Figure 2 ijms-21-07119-f002:**
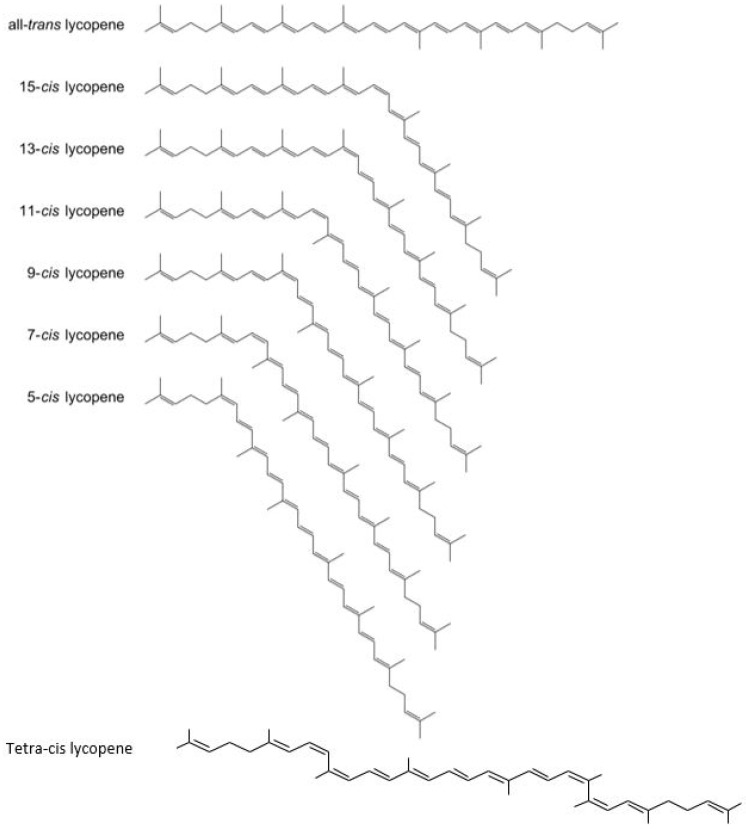
All-*trans*-lycopene and geometrical isomers.

**Figure 3 ijms-21-07119-f003:**
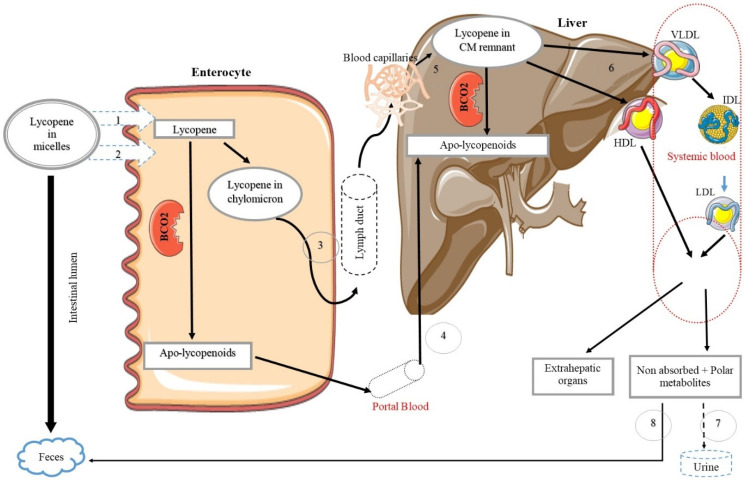
Simplified diagram of lycopene metabolism in the body (reference from [117,119,120,133,134,135,136]). Lycopene enters the enterocytes by active (1) and passive (2) transporters. There, it is packed in chylomicrons or converted to apo-lycopenoids by BCO2. Then, the chylomicrons or apo-lycopenoids are transferred to the liver via the lymphatic (3) and the portal venous (4) systems. Chylomicron remnants (CM) pass to the blood capillaries and are then absorbed by the liver via receptor-mediated endocytosis (5). Lycopene is packaged in very low-density lipoproteins (VLDL) and high-density lipoproteins (HDL) by the liver and released to the systemic circulation (6). Lycopene travels to the extrahepatic organs through the systemic blood and is available there for its biological action. Polar metabolites are excreted in the urine by the kidneys (7), and non-absorbed lycopene is excreted through biliary excretion in feces (8).

**Figure 4 ijms-21-07119-f004:**
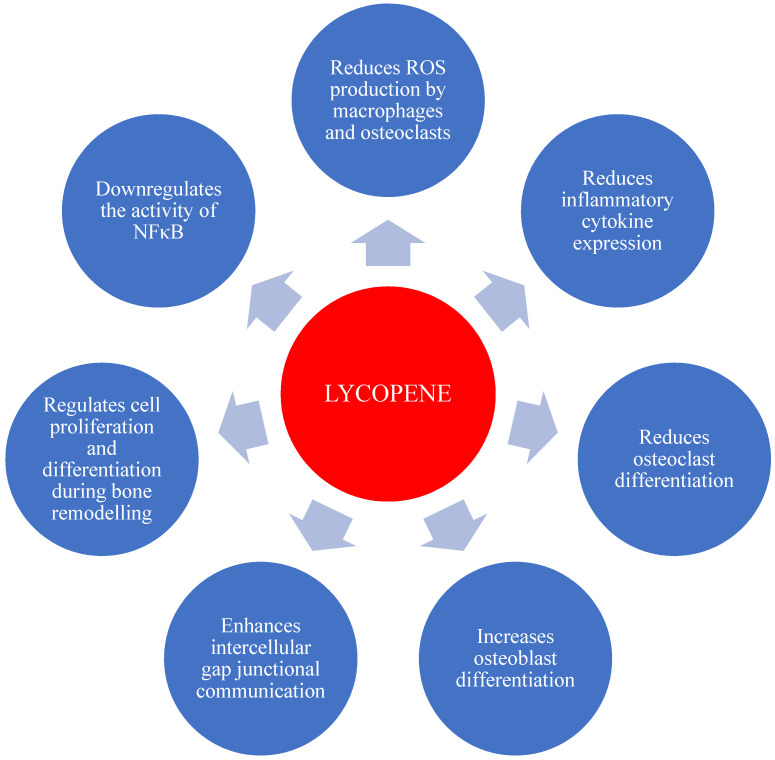
Schematic representation of the potential effects of lycopene on bone cells (reference from [13,14,99].

**Table 1 ijms-21-07119-t001:** Risk factors of postmenopausal osteoporosis.

Fixed Risk Factors	Modifiable Risk Factors
Menopause age [57]	Inadequate calcium and vitamin D intake [58,59]
Menopause and hysterectomy [57,60]	Alcohol consumption [61]
Estrogen deficiency and amenorrhea [62]	Cigarette smoking [63]
Family history of osteoporosis [64,65]	Low body mass index (<20 kg/m^2^) [66]
Previous fractures [67]	Eating disorders [68]
Height loss (>0.5 cm per year) [69]	Inadequate physical exercise [70,71]
Ethnicity (Caucasian and Asian population are at high risk) [72]	Frequent falls [73]

**Table 2 ijms-21-07119-t002:** The effect of lycopene on postmenopausal bone loss based on human trials.

Author and Year	Cohort	Lycopene Formulation and Study Duration	Outcome
Russo et al. (2020) [143]	Postmenopausal women (*n* = 39) Age: 63 ± 7 years	3.9 mg/day as tomato sauce 3 months	Patients who consumed tomato sauce did not show a significant loss of BMD compared to control group
Mackinnon et al. (2011) [141]	Postmenopausal women (*n* = 60) Age: 50–60 years	30 mg/day (regular tomato juice), 70 mg/day (lycopene-rich tomato juice), 30 mg/day (Lyc-O-Mato capsules) 4 months	Lycopene intervention in capsule or juice form supplying at least 30 mg/day led to decreased oxidative stress and bone resorption markers
Mackinnon (2010) [144]	Postmenopausal women (*n* = 45) Age: 55 years	43.33 mg/day supplementation 4 months	Lycopene supplemented group showed significantly lower levels of bone resorption marker (NTx)
Mackinnon et al. (2011) [140]	Postmenopausal women (*n* = 23) Age: 50–60 years	Lycopene intake at baseline and after one month of lycopene restriction was 3.5 mg/d and 0.13 mg/d, respectively (using 7-day dietary records)	Bone resorption marker (NTx) was increased after a month of lycopene restriction Endogenous antioxidant enzymes (SOD and catalase) were decreased after a month of lycopene restriction
Rao et al. (2007) [145]	Postmenopausal women (*n* = 33) Age: 50–60 years	Lycopene intake categorized into four groups as ranged from 1.76 to 7.35 mg/day (using 7-day dietary records)	Serum NTx values dose-dependently decreased Postmenopausal women who consumed 7.35 mg lycopene/day had lower serum NTx compared to the other three groups No difference in bone formation markers

**Table 3 ijms-21-07119-t003:** The effects of lycopene on postmenopausal bone loss based on rodent trials.

Author and Year	Animal Strain	Lycopene Dose and Study Duration	Outcome
Oliveira et al. (2019) [152]	Female Wistar rats	10 mg/kg BW/day 4 weeks pre-OVX and 8 weeks post-OVX	Decreased bone loss in femur epiphysis in the OVX + lycopene group compared to the OVX control group
Li et al. (2018) [153]	Female Sprague-Dawley rats	50 mg/kg BW/day 12 weeks	Higher bone volume and trabecular thickness with low trabecular spaces in the OVX + lycopene group compared to the OVX control group Increased bone contact and bone area around the implant were in the lycopene-treated group compared to controls
Ardawi et al. (2016) [99]	Female Wistar rats	15, 30, 45 mg/kg BW per day 12 weeks	Lycopene treatment dose-dependently enhanced BMD and BMC at the lumbar spine and humerus compared to OVX control group Lycopene (30 and 45 mg/kg BW) increased bone formation markers (serum-OC and serum PINP) while bone resorption markers (serum-CTX-1 and urine-DPD) were decreased
Iimura et al. (2015) [14]	Female Sprague-Dawley 6-week-old	0, 50, 100, 200 mg lycopene/kg diet/day 9 weeks	Lycopene (100 mg/kg) increased lumbar spine BMD and femoral-breaking force compared to OVX control group Bone resorption markers were low in all lycopene-treated groups
Iimura et al. (2014) [13]	Female Sprague–Dawley 6-week-old	0, 50, 100 mg/kg diet lycopene 9 weeks	Lycopene (100 mg/kg) increased BMD of the lumbar spine and the tibial proximal metaphysis compared to OVX control group
Liang et al. (2012) [151]	Female Wistar rats 8-week-old	20, 30, 40 mg/kg BW/day 8 weeks	Lycopene (30 and 40 mg/kg BW) dose-dependently increased BMD and BMC in OVX rats compared to OVX control group

**Table 4 ijms-21-07119-t004:** The effects of lycopene on bone cells (osteoblasts and osteoclasts).

Author and Year	Cell Line	Lycopene Concentration	Outcome
Russo et al. (2020) [143]	Human osteoblast-like cell line Saos-2	5 and 10 μM	Lycopene suppressed RANKL expression indicating the reduction of bone resorption Lycopene reduced the stimulatory effect of ALP within 24 h indicating possible role in mineralization
Oliveira et al. (2019) [152]	Osteoblastic cells from femur medullary canals of ovariectomized female rats	1 μM	Lycopene upregulated the genes associated with bone metabolism of osteoblastic cells within 3–10 days
Costa-Rodrigues et al. (2018) [156]	Osteoblastic cells (human mesenchymal stem cells bone-marrow-derived) Osteoclastic cells (human peripheral blood mononuclear cells)	5 nM−50 μM	Lycopene (≥500 nM) increased osteoblastic cell proliferation and differentiation Lycopene (≥500 nM) significantly decreased osteoclast differentiation
Marcotorchino et al. (2012) [159]	RAW 264.7 cells	0.5, 1, 2 μM	Lycopene dose-dependently reduced the lipopolysaccharides (LPS) mediated activation of inflammatory cytokine (TNF-α) produced by macrophages
Feng et al. (2010) [160]	RAW 264.7 cells	1–10 μM	Lycopene dose-dependently inhibited the increase of nitric oxide production and the secretion of IL-6 when RAW cells were stimulated by LPS
Stefano et al. (2007) [161]	RAW 264.7 cells	5, 10, 20 μM	Lycopene (20 μM) significantly inhibited the ROS accumulated due to addition of gliadin Lycopene (20 μM) significantly inhibited increase in nitric oxide synthase levels
Rao et al. (2003) [34]	Osteoclast were generated from bone marrow cells	0.01, 0.1, 1, 10 μM	Lycopene (10 μM) significantly inhibited PTH stimulated resorption by osteoclasts

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
