# Peer review of "Potential Role of Lycopene in the Prevention of Postmenopausal Bone Loss: Evidence from Molecular to Clinical Studies"

_ijms, 2020, doi:10.3390/ijms21197119_

Round 1

Reviewer 1 Report

The purpose of this paper is to evaluate the Potential role of lycopene in the prevention of postmenopausal bone loss taking into account evidence from human, animal and cellular studies. 

This review, is overall interesting and susceptible to publication after revision. However, some considerations should be taken into account:

-Lane 167 "longer sleep duration decreases the BMD in elderly Korean people". Please could you better explain this sentence?

-Lane 182-185 "Interestingly, another study showed that long term estrogen deficiency increased both bone formation and resorption markers in postmenauposal......".

These evidences are very interesting, could you explain or hypothesize

the mechanisms at the basis of these apparently contrasting processes?

_The authors reported that lycopene has a protective effect against bone loss  in vitro, in animal models of osteoporosis, and in human clinical studies but they do not draw conclusions about the use of this molecule to prevent osteoporosis in humans, and on its possible dosage. Please discuss this point and add in the Conclusions Section, the desirable future use of lycopene supplementation in the prevention of osteoporosis.

Author Response

The authors thank the reviewers for their time, commitment, and helpful comments.  The reviewer's concerns are addressed individually below.

Comment 1 (x) Extensive editing of English language and style required

Response 1: Errors in English spelling, grammar, punctuation, and sentence structure have been edited by a native English speaker.

Comment 2: Line 167 "longer sleep duration decreases the BMD in elderly Korean people". Please could you better explain this sentence?

Response 2: Following the reviewer’s comment, we revised the text accordingly. Please see lines 181-189 of the revised manuscript.

“Interestingly, some cross-sectional studies have found an inverse relationship between sleep duration and BMD in elderly women [60,61]. It has been hypothesised that a shorter wake time reduces the daily mechanical loading that induces bone remodelling, and thus reduces BMD [61]. Lower melatonin levels due to reduced light exposure may lead to fewer interactions between estrogen and its receptors and thus negatively impact BMD [62]. However, other studies report either positive or null relationships between sleep duration and BMD in postmenopausal women [63-66]. The conflicting evidence in this area may be due to confounding study participant factors that are not uniformly controlled for across the studies such as age, body composition, diet, and nighttime-only sleep duration versus inclusion of daytime naps.”

Comment 3: Lines 182-185 "Interestingly, another study showed that long term oestrogen deficiency increased both bone formation and resorption markers in postmenopausal......". These evidences are very interesting, could you explain or hypothesize the mechanisms at the basis of these apparently contrasting processes?

Response 3: Following the reviewer’s comment, we revised the text accordingly. Please see lines 203-214 of the revised manuscript.

Interestingly, studies have shown that long term estrogen deficiency increased both bone resorption and bone formation markers in postmenopausal women, suggestive of enhanced bone turnover with increased net bone loss [87-90]. Estrogen deficiency increases renal calcium excretion while decreasing intestinal calcium absorption [10], and the resultant fall in calcium levels can activate various bone resorption mechanisms that include PTH, osteocalcin, OPG, and the RANK/RANKL system [90-94]. These bone resorption markers are therefore found in the blood in higher concentrations in osteoporosis. Conversely, osteocalcin, which is secreted by osteoclasts, directly binds calcium and enables bone mineralization by increasing hydroxyapatite absorption, and thus is a marker of bone formation [95]. But insufficient calcium and phosphorous stores in osteoporotic women reduce hydroxyapatite crystals formation, leaving more osteocalcin free to circulate in the blood [92,93]. The molecular mechanisms responsible for these complex changes are not yet fully elucidated [87].

Comment 4: The authors reported that lycopene has a protective effect against bone loss in vitro, in animal models of osteoporosis, and in human clinical studies but they do not draw conclusions about the use of this molecule to prevent osteoporosis in humans, and on its possible dosage. Please discuss this point and add in the Conclusions Section, the desirable future use of lycopene supplementation in the prevention of osteoporosis.

Response 4: We have revised the conclusions section accordingly. Please see lines 494-507 of the revised manuscript.

Postmenopausal bone loss is a public health issue in the ageing population. Medications can prevent and treat bone loss in some patients but may have negative side effects. Phytochemicals present in fruits and vegetables play a major role in non-communicable disease management. As summarized in this review, lycopene has a protective effect against bone loss; this has been demonstrated in vitro, in animal models of osteoporosis, and in human clinical studies. The epidemiological and clinical studies discussed in this review have demonstrated that lycopene intake (>30 mg/day) was effective in reducing bone resorption markers in postmenopausal women. Tomatoes are a major source of dietary lycopene; however, due to relatively low bioavailability of all-trans lycopene isomers found in red tomatoes, several techniques are being used to increase bioavailability such as adding oils or making cooked sauces and pastes. Other sources of more bioavailable lycopene can be found in nature. For example, orange heirloom tomatoes contain >90% of lycopene in a cis isomeric form which is estimated to be 8.5 times more bioavailable than all-trans lycopene [16]. Further studies are warranted to compare the relative benefits of these tomato varieties and lycopene isomers in protecting against bone loss.

Reviewer 2 Report

3.1 Risk factors of postmenopausal osteoporosis - please be more specific, which mutation of genes are responsible for hereditary osteoporosis? 

Author Response

The authors thank the reviewers for their time, commitment, and helpful comments.  The reviewer's concerns are addressed individually below.

Comment 1: 3.1 Risk factors of postmenopausal osteoporosis - please be more specific, which mutation of genes are responsible for hereditary osteoporosis? 

Response 1:

Following the reviewer’s comment, we have revised the section accordingly. Please see lines 164-171 of the revised manuscript.

“Heredity is the major non-modifiable factor of osteoporosis, and children of parents with osteoporosis and fractures are themselves more prone to develop osteoporosis [50]. Osteoporosis is a polygenic disease which involve several genes. In rare instances, osteoporosis can be inherited due to mutations in single genes. Mutations of two genes of type 1 collagen (COL1A1 and COL1A2) are responsible for the dominant osteoporotic disease called “osteogenesis imperfecta” which is characterized by low bone mass and increased bone fragility [51-54]. Osteoporosis inheritance has also been linked with inactivating mutations in the aromatase (CYP19A1) and estrogen receptor alpha genes (ERα) [55,56].”